# Discovery of Novel Benzamide-Based Sigma-1 Receptor Agonists with Enhanced Selectivity and Safety

**DOI:** 10.3390/molecules30173584

**Published:** 2025-09-02

**Authors:** Pascal Carato, Bénédicte Oxombre, Séverine Ravez, Rajaa Boulahjar, Marion Donnier-Maréchal, Amélie Barczyk, Maxime Liberelle, Patrick Vermersch, Patricia Melnyk

**Affiliations:** 1Univ. Lille, Inserm, CHU Lille, U1172-LilNCog-Lille Neuroscience & Cognition, F-59000 Lille, France; pascal.carato@univ-poitiers.fr (P.C.); benedicte.vanteghem@univ-lille.fr (B.O.); severine.ravez@univ-lille.fr (S.R.); maxime.liberelle@univ-lille.fr (M.L.); patrick.vermersch@univ-lille.fr (P.V.); 2Univ. Lille, Inserm, CHU Lille, U1286-INFINITE-Institute for Translational Research in Inflammation, F-59000 Lille, France; amelie.barczyk@univ-lille.fr

**Keywords:** Sigma-1 receptor, central nervous system (CNS), benzamide derivatives, pharmacomodulation, ADME, drug discovery

## Abstract

Central nervous system (CNS) disorders such as neurodegenerative diseases, multiple sclerosis, or even brain ischemia represent major therapeutic challenges with limited effective treatments. The sigma-1 receptor (S1R), a unique ligand-operated molecular chaperone enriched at mitochondria-associated membranes, has emerged as a promising drug target due to its role in neuroprotection and neuroinflammation. Building upon our previously identified S1R ligand (compound **1**), we designed and synthesized six novel benzamide derivatives through pharmacomodulation to optimize affinity, selectivity, and safety profiles. Among these, compound **2** demonstrated superior S1R affinity, improved selectivity over the sigma-2 receptor (S2R), and favorable ADME properties, including enhanced permeability and markedly reduced in vitro cardiac toxicity compared to the lead compound. Functional assays confirmed the agonist activity of key derivatives, while safety evaluations revealed low cytotoxicity and minimal off-target receptor interactions. Collectively, these findings support compound **2** as a promising candidate for further preclinical development in S1R-related CNS disorders.

## 1. Introduction

Neurodegenerative diseases and ischemic stroke represent major health challenges of the 21st century, contributing to rising morbidity and mortality through progressive physical and cognitive decline. A key pathological hallmark shared by these conditions is the disruption of mitochondria-endoplasmic reticulum (ER) communication [1,2]. This specialized subcellular interface, known as the mitochondria-associated membrane (MAM), plays a central role in maintaining cellular homeostasis and is critically involved in diverse biological processes, including calcium signaling, lipid metabolism, inflammation, autophagy, apoptosis, and ER stress responses [3].

The MAM harbors a variety of functional proteins, among which the sigma-1 receptor (S1R) has attracted increasing interest over the past decade due to its emerging role in brain disorders [4,5,6]. S1R is a highly conserved and dynamic protein expressed across species and cell types. Although present in the ER, nuclear, and plasma membranes, it is particularly enriched at the MAM. In the central nervous system (CNS), S1R is widely expressed in both neurons and glial cells, where it modulates key physiological functions such as neuroprotection, neuroinflammation, neurotransmission, and neuroplasticity.

Unlike classical receptors, S1R does not conform to standard pharmacological paradigms: its endogenous ligand remains elusive, and no canonical signal transduction cascade has been identified. Instead, S1R functions as a unique, ligand-operated molecular chaperone. Upon specific activation, S1R translocates to multiple subcellular compartments—including the ER, nuclear, mitochondrial, and plasma membranes, as well as cytosolic vesicles and even the extracellular milieu—where it modulates the activity of various receptors, ion channels, transporters, and enzymes, either enhancing or attenuating their signaling.

Given its multifaceted role, S1R has emerged as a promising therapeutic target. Several small molecules with favorable pharmacokinetic and pharmacodynamic properties have been developed, both as potential drugs and as molecular imaging tracers [7]. For instance, blarcamesine (Anavex 2-73), a compound targeting S1R and muscarinic receptors, has successfully completed Phase 2/3 clinical trials for Rett syndrome and early-stage Alzheimer’s disease (AD) [8,9]. Pridopidine, which targets S1R and dopamine D2/D3 receptors, has undergone Phase 2 evaluation for levodopa-induced dyskinesia and Phase 3 trials for early Huntington’s disease (HD) [10,11].

Our team previously identified the tetrahydroisoquinoline-hydantoin scaffold as a key pharmacophore for S1R modulation [12]. Compounds selected based on their S1R agonist profiles have demonstrated promising potential in preclinical models of central nervous system (CNS) disorders, including brain ischemia, multiple sclerosis (MS), and cocaine addiction [13,14]. Through pharmacomodulation efforts, we synthesized several series of compounds featuring benzannulated motifs, tricyclic heterocycles, and benzamide scaffolds, all exhibiting high affinity for S1R [15]. These compounds typically consist of a heterocyclic core linked to a benzylmethylamine moiety via aliphatic chains of two, three, or four methylene units. Among the various series, the simple benzamide derivatives stood out for their improved metabolic stability, excellent S1R-binding affinity, high selectivity, and lack of in vitro cytotoxicity [15]. Comprehensive evaluation of absorption, distribution, metabolism, and excretion (ADME) properties, along with safety pharmacology and toxicology studies, was conducted using standardized in vitro and in vivo methodologies. These assessments confirmed favorable drug-like characteristics, including the potential for oral administration, supporting their development as selective and safe S1R-targeted therapeutic candidates [16].

Building on these promising findings, further development of next-generation S1R ligands is warranted to enhance efficacy, selectivity, and drug-like properties. Starting from the previously identified benzamide scaffold and its most potent derivative (compound **1**), we designed a new series of analogs by inverting the amide bond while maintaining the distance between the two phenyl rings (Figure 1). Structural variations were introduced by modulating both the position and the nature of the substituent (R group), incorporating halogen atoms (Cl, Br) as well as electron-withdrawing groups such as cyano and nitro moieties (compounds **2**–**7**).

## 2. Results and Discussion

### 2.1. Chemistry

Starting from the commercial substituted *N*-benzylamine (**8a**–**f**), reaction in dichloromethane with 4-chloropropionyl chloride gave the corresponding compounds **9a**–**f**. These derivatives **9a**–**f** were engaged in benzylmethylamine, which reacted both as a base and a solvent, and afforded the final compounds **2**–**7** (Figure 1, Appendix A).

### 2.2. Biological Evaluation and SAR

S1R and S2R affinities were classically evaluated through competitive binding assays using radioligands, following the protocol described by Ganapathy et al. [17]. The selective radioligand [^3^H] (+)-pentazocine was employed as the classic prototypical ligand, selectively binding to the orthosteric S1R-binding sites. [^3^H]-DTG was used as the most useful radioligand to study compounds classified as S2R ligands, in the presence of an excess of unlabeled (+)-pentazocine to selectively block S1R. Jurkat cell membranes served as the receptor source in both assays. The equilibrium dissociation constants (*K*i) for S1R and S2R were derived from the IC_50_ values of each compound, and selectivity ratios (S2R/S1R) were subsequently calculated (Table 1, Appendix A).

The lead compound **1** exhibited high affinity for S1R (*K*i = 3.2 nM), with a selectivity ratio of 190. Substitution with a chloro atom at the *para* position (compound **3**) improved the S1R affinity (*K*i = 1.7 nM) and selectivity (S2R/S1R = 241). The highest S1R affinity (*K*i = 0.6 nM) was achieved with a chloro substituent at the *meta* position (compound **2**), displaying excellent selectivity (S2R/S1R = 317). In contrast, introducing a bulkier bromo atom at the *meta* position (compound **4**) drastically reduced the S1R affinity (*K*i > 200 nM). The dichloro-substituted derivative **5** maintained good S1R affinity (*K*i = 2.3 nM) but showed reduced selectivity (S2R/S1R = 52). Electron-withdrawing cyano substitution (compound **6**) preserved good S1R affinity (*K*i = 5.6 nM) and selectivity (S2R/S1R = 331), comparable to compound **3**. Notably, compound **7,** with a withdrawing nitro substituent, showed a lower S1R affinity (*K*i = 110 nM) together with a lower selectivity ratio (S2R/S1R = 59). Chloro and cyano compounds (**2**, **3,** and **6**) demonstrated excellent selectivity (S2R/S1R = 241–331), with strong S1R affinities (*K*i = 0.6–5.6 nM).

To assess potential cytotoxicity in CNS-relevant cells, human neuroblastoma (SH-SY5Y) cells were exposed to compounds **2**–**7** at concentrations up to 100 μM. Cell viability was measured using the MTT assay. All compounds displayed minimal cytotoxicity, with IC_50_ values exceeding 100 μM and selectivity indices (IC_50_(SH-SY5Y)/*K*i(S1R)) ranging from 909 to 166,666 (Table 1). In particular, compound **2**, combining high S1R affinity (*K*i = 0.6 nM) and selectivity (317), achieved the highest selectivity index of 166,666.

### 2.3. Docking Study of Compounds

To investigate the potential binding modes of these benzamide derivatives, compounds **1** and **2** were docked into the binding site of S1R [18]. As illustrated in Figure 2, the retroamide modification in compound **2** resulted in a slight shift of the compound inside the binding pocket. This conformational shift enhanced the ionic interaction with Asp126 while preserving the key salt bridge with Glu172. Additionally, the new orientation enabled the formation of an extra hydrogen bond with the phenolic group of Tyr103, potentially contributing to the improved affinity observed for compound **2**. In contrast, the docking study did not provide a rationale for the pronounced loss of affinity observed with the brominated derivative **4**.

### 2.4. Off-Target Analysis over a Panel of 58 Recombinant Human Receptors

Among the synthesized compounds, four derivatives were selected based on their affinity and selectivity profiles. Compound **5**, exhibiting the lowest S2R/S1R selectivity [17], and compound **4**, characterized by poor S1R affinity (*K*i > 200 nM), were excluded from further evaluation. The remaining compounds (**2**, **3**, **6**, and **7**) were assessed for their agonist and antagonist activities across a panel of 58 receptors relevant to CNS disorders, including serotonin, adenosine, adrenergic, cannabinoid, dopamine, histamine, and muscarinic receptors. These assays were conducted at a concentration of 10 µM, in duplicate, utilizing aequorin [19], cAMP HTRF™ [20], and GTPγS binding assays [21], as detailed in Table 2. None of the compounds exhibited significant agonist activity at 10 µM, except for compound **3**, which showed moderate agonist activity at the 5-HT_7_ serotonin receptor. In contrast, all tested compounds demonstrated antagonist activity at 10 µM against adrenergic α_1_ and histamine H_1_ receptors. Notably, the reference compound **1** showed no activity at the H_1_ receptor. Compounds **2** and **3**, bearing chloro substituents at the *meta* and *para* positions of the benzyl ring, respectively, exhibited antagonist activity against eight and ten receptors, respectively. Conversely, compounds **6** and **7**, substituted with electron-withdrawing cyano and nitro groups, respectively, displayed antagonist activity at a limited number of receptors. Specifically, compound **6** antagonized four receptors (5-HT_2A_, 5-HT_2C_, α_1a_, H_1_), while compound **7** was active against five receptors (5-HT_2A_, 5-HT_3_, α_1a_, H_1_, M_1_). Both compounds **6** and **7** demonstrated profiles similar to the lead compound **1**, with antagonist activity targeting three adrenergic and muscarinic receptors (α_1a_, α_1b_, and M_1_).

### 2.5. Target Engagement and Agonist Profile of Selected Compounds

The interaction between S1R and BiP provides a reliable assay to identify S1R ligands by monitoring the dynamics of the S1R-BiP complex [22,23,24]. In this assay, the known S1R agonist PRE-084 and antagonist NE-100 were used as controls (Figure 3). As shown in Figure 3, PRE-084 induced a rapid and transient dissociation of the S1R-BiP heterodimer (reducing complex levels to less than 50% of the control), whereas NE-100 prevented this dissociation, maintaining complex levels at 90–100% of the control, both in the presence and absence of PRE-084. In the present study, the most potent and selective compounds (**2**, **3**, and **6**) were evaluated using this assay. Compounds **2** and **3**, at concentrations of 1 and 10 µM, induced a significant, dose-dependent dissociation of the S1R-BiP complex. In contrast, compound **6** produced only a slight and less significant reduction. The dissociation induced by compounds **2** and **3** was completely inhibited by NE-100 (10 µM), confirming their agonist activity. Moreover, co-treatment with PRE-084 and the test compounds resulted in dissociation levels comparable to PRE-084 alone, indicating that these compounds do not antagonize PRE-084’s effect, thus validating the absence of the antagonist PRE-084.

### 2.6. ADME Studies of Selected Compounds

Based on activity, selectivity, and agonist profiles, compounds **2** and **6** were selected for advanced stages of drug development. The most promising candidate (compound **2**) bears a chloro substituent, while compound **6** is a cyano-substituted derivative.

Preliminary ADME evaluation focused on bioavailability properties at 10 µM (Table 3). Both compounds exhibited high solubility in PBS at pH 7.4 [25]. Solubility and stability remained unchanged in simulated gastric fluid (SGF) and simulated intestinal fluid (SIF), which mimic gastrointestinal digestion. Intestinal permeability was assessed using a two-chamber system with apical (A, pH 6.5) and basolateral (B, pH 7.4) compartments, representing the luminal and blood/mesenteric lymph sides of the gastrointestinal tract, respectively [26]. Both compounds demonstrated good A-to-B permeability, with compound **6** showing higher permeability than compound **2**. The efflux ratios for both compounds ranged between 1 and 2, indicating improved permeability relative to the reference compound **1**. Both compounds displayed plasma stability exceeding 24 h. Plasma protein binding was high for compound **2** (98%), comparable to that of compound **1**, while compound **6** exhibited moderate binding (63%) [27].

Microsomal stability was evaluated using human liver microsomes at 10 µM (Table 3). Compound **6** exhibited a half-life exceeding 1 h and favorable clearance (<115 µL/min/mg protein). In contrast, compound **2** had a shorter half-life (5 min) and higher clearance (135.3 µL/min/mg protein). Since cytochrome P450 (CYP) enzymes—especially families 1 to 3—play major roles in drug metabolism [28], CYP inhibition was tested against the human isoforms CYP1A, 2B6, 2C8, 2C9, 2C19, 2D6, and 3A, as recommended by FDA and EMA guidelines. Both compounds exhibited no or weak inhibition, except for moderate inhibition of CYP2D6. Membrane transport proteins, including the ATP-binding cassette (ABC) and solute carrier (SLC) families, affect drug pharmacokinetics and potential drug–drug interactions [29,30,31,32,33,34,35,36,37,38]. The inhibitory effects of compounds **2** and **6** were assessed against five ABC transporters (P-gp, BCRP, MRP1, MRP2, and MRP3) and seven SLC transporters (OAT1, OAT3, OATP1B1, OCT1, OCT2, ASBT, and NTCP). Both compounds demonstrated low inhibitory activity (<50% inhibition), comparable to that of compound **1**, indicating negligible transporter inhibition.

Cardiac toxicity is a leading cause of drug development failure. Early safety profiling includes evaluation of the potassium voltage-gated channel subfamily H member 2 (KCNH2 or hERG) channel, the only in vitro assay mandated by regulatory agencies to predict proarrhythmic risk [29]. Compounds **2** and **6** inhibited hERG currents with IC_50_ values in the micromolar range. Importantly, an IC_50_ (hERG)/*Ki* (σ_1_) ratio exceeding 100 is advised for cardiac safety; all compounds met this criterion. Compound **2** was particularly notable, with an IC_50_ (hERG)/*Ki* (σ_1_) ratio above 11,000, suggesting a high potential for cardiac safety.

## 3. Materials and Methods

### 3.1. Chemistry

All chemicals and solvents were obtained from commercial suppliers and, unless stated otherwise, were used without further purification. Reaction progress was monitored by thin-layer chromatography (TLC) on Macherey-Nagel Alugram^®^ Sil 60/UV254 plates (0.2 mm; Macherey-Nagel GmbH & Co. KG, Düren, Germany). Products were purified by column chromatography on Macherey-Nagel silica gel (230–400 mesh).

The ^1^H and ^13^C NMR spectra were recorded on a Bruker DRX 300 spectrometer (300 and 75 MHz, respectively; Division Biospin, Wissembourg, France). Chemical shifts are reported in ppm relative to tetramethylsilane (TMS) and described as δ (ppm), multiplicity (s, d, t, q, p, dd, br, m), coupling constant (*J*, Hz), integration, and assignment.

Mass spectra for compounds **9a**–**f** and **2**–**7** were acquired with unit mass accuracy using an LCMS system (Waters Alliance Micromass ZQ 2000; Waters Corporation, Milford, MA, USA) equipped with PDA detection, an electrospray ionization (ESI) source, and a Waters XBridge C18 column (5 μm, 50 × 4.6 mm). The mobile phase consisted of a 4 min gradient from 98% H_2_O/formate buffer (5 mM, pH 3.8) to 100% CH_3_CN/formate buffer (5 mM, pH 3.8) at a flow rate of 2 mL/min, followed by 1 min re-equilibration.

**General Procedure for the Preparation of 3-Chloro-*N*-(4-substitutedbenzyl)propanamide (9a–f):** To a solution of 4-substituted benzylamine **8a**–**f** (7.0 mmol) in 15 mL of DCM at 0 °C, a solution of 3-chloropropionyl chloride (7.0 mmol, 668 µL) was slowly added in 5 mL of DCM. The resulting mixture was stirred at room temperature for 12 h. Then, the reaction was quenched with 20 mL of water, and the product was extracted with 3 × 25 mL of DCM. The combined organic fractions were dried over magnesium sulfate. The crude product was purified by column chromatography.

***N*-(3-Chlorobenzyl)-3-chloropropanamide (9a):** Yield: 73%. *R*f (DCM: MeOH (saturated with gaz NH_3_) 9.5/0.5) = 0.7. ^1^H NMR (300 MHz, CDCl_3_), δ: 7.40–7.10 (m, 4H, H_aro_); 6.10 (br s, 1H, NH); 4.43 (d, *J* = 6.2 Hz, 2H, CH_2_); 3.80 (t, *J* = 6.4 Hz, 2H, CH_2_); 2.65 (t, *J* = 6.3 Hz, 2H, CH_2_). ^13^C NMR (75 MHz, CDCl_3_) δ: 169.6 (CO); 139.9 (C_aro_); 134.5 (C_aro_); 130.0 (C_aro_); 137.7 (C_aro_); 125.8 (C_aro_); 43.1 (CH_2_); 40.1 (CH_2_); 39.5 (CH_2_). LCMS *m*/*z* calc for C_10_H_12_Cl_2_NO [M + H]^+^: 232.0, 234.0, 236.0; found: 231.9, 233.9, 235.9.

***N*-(4-Chlorobenzyl)-3-chloropropanamide (9b):** Yield: 60%. *R*f (ethyl acetate: cyclohexane, 5:5) = 0.7. ^1^H NMR (300 MHz, CDCl_3_), δ: 7.35–7.30 (m, 2H, H_aro_); 7.25–7.20 (m, 2H, H_aro_); 5.94 (br s, 1H, NH); 4.45 (d, *J* = 6.2 Hz, 2H, CH_2_); 3.84 (t, *J* = 6.4 Hz, 2H, CH_2_); 2.67 (t, *J* = 6.3 Hz, 2H, CH_2_). ^13^C NMR (75 MHz, CDCl_3_) δ: 169.4 (CO); 136.4 (C_aro_); 133.4 (C_aro_); 129.1 (C_aro_); 128.9 (C_aro_); 43.1 (CH_2_); 40.1 (CH_2_); 39.6 (CH_2_). LCMS *m*/*z* calc for C_10_H_12_Cl_2_NO [M + H]^+^: 232.0, 234.0, 236.0; found: 232.1, 234.1, 236.1.

***N*-(4-Bromobenzyl)-3-chloropropanamide (9c):** Yield: 67%. *R*f (DCM: MeOH (saturated with gaz NH_3_) 9.5/0.5) = 0.7. ^1^H NMR (300 MHz, CDCl_3_), δ: 7.50–7.40 (m, 2H, H_aro_); 7.30–7.15 (m, 2H, H_aro_); 6.10 (br s, 1H, NH); 4.45 (d, *J* = 5.9 Hz, 2H, CH_2_); 3.83 (t, *J* = 6.4 Hz, 2H, CH_2_); 2.68 (t, *J* = 6.4 Hz, 2H, CH_2_). ^13^C NMR (75 MHz, CDCl_3_) δ: 169.5 (CO); 140.2 (C_aro_); 130.7 (C_aro_); 130.6 (C_aro_); 130.3 (C_aro_); 126.3 (C_aro_); 122.7 (C_aro_); 43.1 (CH_2_); 40.1 (CH_2_); 39.5 (CH_2_). LCMS *m*/*z* calc for C_10_H_12_BrClNO [M + H]^+^: 275.9, 277.9, 279.9; found: 275.8, 277.8, 279.8.

**3-Chloro*-N*-(2,4-dichlorobenzyl)propanamide (9d):** Yield: 79%. *R*f (DCM: MeOH (saturated with gaz NH_3_) 9.5/0.5) = 0.6. ^1^H NMR (300 MHz, CDCl_3_), δ: 7.40 (d, *J* = 2.0 Hz, 1H, H_aro_); 7.36 (d, *J* = 8.2 Hz, 1H, H_aro_); 7.23 (dd, *J* = 8.2 Hz, *J* = 2.1 Hz, 1H, H_aro_); 6.05 (br s, 1H, NH); 4.53 (d, *J* = 6.1 Hz, 2H, CH_2_); 3.83 (t, *J* = 6.4 Hz, 2H, CH_2_); 2.68 (t, *J* = 6.4 Hz, 2H, CH_2_). ^13^C NMR (75 MHz, CDCl_3_) δ: 169.6 (CO); 134.1 (C_aro_); 134.0 (C_aro_); 133.9 (C_aro_); 130.8 (C_aro_); 129.3 (C_aro_); 127.4 (C_aro_); 41.0 (CH_2_); 40.0 (CH_2_); 39.4 (CH_2_). LCMS *m*/*z* calc for C_10_H_10_Cl_3_NO [M − H]^−^: 264.0, 266.0, 268.0, 270.0; found: 263.8, 265.8, 267.9, 269.9.

**3-Chloro*-N*-(4-cyanobenzyl)propanamide (9e):** Yield: 65%. *R*f (DCM: MeOH (saturated with gaz NH_3_) 9.5/0.5) = 0.6. ^1^H NMR (300 MHz, CDCl_3_), δ: 7.62 (d, *J* = 8.5 Hz, 2H, H_aro_); 7.48 (d, *J* = 8.1 Hz, 2H, H_aro_); 6.05 (br s, 1H, NH); 4.55 (d, *J* = 6.1 Hz, 2H, CH_2_); 3.85 (t, *J* = 6.3 Hz, 2H, CH_2_); 2.70 (t, *J* = 6.3 Hz, 2H, CH_2_). ^13^C NMR (75 MHz, CDCl_3_) δ: 169.9 (CO); 143.7 (C_aro_); 132.4 (C_aro_); 128.1 (C_aro_); 118.7 (C_aro_); 111.1 (C_aro_); 43.1 (CH_2_); 40.1 (CH_2_); 39.4 (CH_2_). LCMS *m*/*z* calc for C_11_H_10_ClN_2_O [M − H]^−^: 221.0, 223.0; found: 220.9, 222.9.

**3-Chloro*-N*-(4-nitrobenzyl)propanamide (9f):** Yield: 69%. *R*f (DCM: MeOH (saturated with gaz NH_3_) 9.5/0.5) = 0.6. ^1^H NMR (300 MHz, CDCl_3_), δ: 8.20 (d, *J* = 8.8 Hz, 2H, H_aro_); 7.48 (d, *J* = 8.9 Hz, 2H, H_aro_); 6.05 (br s, 1H, NH); 4.60 (d, *J* = 6.2 Hz, 2H, CH_2_); 3.85 (t, *J* = 6.3 Hz, 2H, CH_2_); 2.70 (t, *J* = 6.2 Hz, 2H, CH_2_). ^13^C NMR (75 MHz, CDCl_3_) δ: 169.7 (CO); 145.5 (C_aro_); 128.2 (C_aro_); 124.0 (C_aro_); 123.9 (C_aro_); 43.0 (CH_2_); 40.1 (CH_2_); 39.5 (CH_2_). LCMS *m*/*z* calc for C_10_H_10_ClN_2_O_3_ [M − H]^−^: 241.0, 243.0; found: 240.8, 242.8.

**General Procedure for the Preparation of 3-(Benzylmethylamino)-*N*-(substitutedbenzyl)propanamide (2-7)**: A solution of compounds **9a**–**f** (1 eq) in *N*,*N*-benzylmethylamine (6 eq) was stirred at room temperature for 18 h. A 5 mL amount of cyclohexane was added, and the white solid was filtered. The filtrate was concentrated and purified by thick-layer chromatography or column chromatography.

**3-(Benzylmethylamino)-*N*-(3-chlorobenzyl)propanamide (2):** Compound **2** was synthesized by using *N*-(3-chlorobenzyl)-3-chloropropanamide **9a** (0.43 mmol, 100 mg) and *N*-benzylmethylamine (4.3 mmol, 555 µL). Purification by thick-layer chromatography (DCM: MeOH(NH_3_), 9.5:0.5, *R*f = 0.3). Yield: 68%. ^1^H NMR (300 MHz, CDCl_3_), δ: 8.80 (br s, 1H, NH); 7.30–7.00 (m, 9H, H_aro_); 4.40 (d, *J* = 5.6 Hz, 2H, CH_2_); 3.50 (s, 2H, CH_2_); 2.70 (t, *J* = 6.0 Hz, 2H, CH_2_); 2.48 (t, *J* = 5.6 Hz, 2H, CH_2_); 2.20 (s, 3H, CH_3_). ^13^C NMR (75 MHz, CDCl_3_) δ: 172.5 (CO); 140.8 (C_aro_); 137.5 (C_aro_); 134.4 (C_aro_); 129.9 (C_aro_); 129.0 (C_aro_); 128.5 (C_aro_); 127.8 (C_aro_); 127.5 (C_aro_); 127.4 (C_aro_); 125.9 (C_aro_); 62.2 (CH_2_); 53.1 (CH_2_); 42.5 (CH_3_); 40.1 (CH_2_); 32.6 (CH_2_). LCMS *m*/*z* calc for C_18_H_22_ClN_2_O [M + H]^+^: 317.1, 319.1; found: 317.1, 319.1.

**3-(Benzylmethylamino)-*N*-(4-chlorobenzyl)propanamide (3):** Compound **3** was synthesized by using *N*-(4-chlorobenzyl)-3-chloropropanamide **9b** (0.21 mmol, 50 mg) and *N*,*N*-benzylmethylamine (1.29 mmol, 166 µL). Purification by thick-layer chromatography with cyclohexane/ethyl acetate/MeOH (saturated with gas NH_3_), 4.5:4.5:1, *R*f = 0.4. Yield: 67%. ^1^H NMR (300 MHz, CDCl_3_), δ: 8.65 (br s, 1H, NH); 7.34–7.19 (m, 7H, H_2_, H_6_, H_aro_); 7.09 (d, *J* = 9.3 Hz, 2H, H_3_, H_5_); 4.38 (d, *J* = 6.1 Hz, 2H, CH_2_); 3.57 (s, 2H, CH_2_); 2.79 (t, *J* = 6.2 Hz, 2H, CH_2_); 2.56 (t, *J* = 6.0 Hz, 2H, CH_2_); 2.25 (s, 3H, CH_3_). ^13^C NMR (75 MHz, CDCl_3_) δ: 171.9 (CO); 137.2 (C_aro_); 133.1 (C_aro_); 129.3 (2 C_aro_); 129.2 (3 C_aro_); 128.8 (2 C_aro_); 128.6 (2 C_aro_); 127.9 (C_aro_); 62.0 (CH_2_); 53.0 (CH_2_); 42.6 (CH_2_); 40.7 (CH_3_); 32.4 (CH_2_). LCMS *m*/*z* calc for C_18_H_22_ClN_2_O [M + H]^+^: 317.1, 319.1; found: 316.9, 319.0.

**3-(Benzylmethylamino)-*N*-(3-bromobenzyl)propanamide (4):** Compound **4** was synthesized by using *N*-(4-bromobenzyl)-3-chloropropanamide **9c** (0.21 mmol, mg) and *N*,*N*-benzylmethylamine (1.29 mmol, 166 µL). Purification by thick-layer chromatography (DCM/MeOH (saturated with gas NH_3_) 9.5/0.5, *R*f = 0.3). Yield: 74%. %. ^1^H NMR (300 MHz, CDCl_3_), δ: 8.84 (br s, 1H, NH); 7.43–7.40 (m, 2H, H_aro_); 7.27–7.17 (m, 5H, H_aro_); 7.06–7.02 (m, 2H, H_aro_); 4.39 (d, *J* = 5.7 Hz, 2H, CH_2_); 3.48 (s, 2H, CH_2_); 2.68 (t, *J* = 5.2 Hz, 2H, CH_2_); 2.48 (t, *J* = 5.3 Hz, 2H, CH_2_); 2.20 (s, 3H, CH_3_). ^13^C NMR (75 MHz, CDCl_3_) δ: 172.4 (CO); 141.2 (C_aro_); 137.6 (C_aro_); 130.7 (C_aro_); 130.4 (C_aro_); 130.2 (C_aro_); 130.0 (C_aro_); 128.5 (C_aro_); 127.5 (C_aro_); 126.4 (C_aro_); 122.7 (C_aro_); 62.3 (CH_2_); 53.1 (CH_2_); 42.5 (CH_3_); 41.0 (CH_2_); 32.6 (CH_2_). LCMS *m*/*z* calc for C_18_H_22_BrN_2_O [M + H]^+^: 361.1, 363.1; found: 361.0, 363.0.

**3-(Benzylmethylamino)-*N*-(2,4-Dichlorobenzyl)propanamide (5):** Compound **5** was synthesized by using 3-chloro-*N*-(2,4-dichlorobenzyl)propanamide **9d** (0.37 mmol, 100 mg) and *N*-benzylmethylamine (3.7 mmol, 483 µL). Purification by thick-layer chromatography (DCM: MeOH(NH_3_), 9.7:0.3, *R*f = 0.3). Yield: 80%. ^1^H NMR (300 MHz, CDCl_3_), δ: 8.90 (br s, 1H, NH); 7.38 (d, *J* = 1.9 Hz, 1H, H_aro_); 7.28 (d, *J* = 8.3 Hz, 1H, H_aro_); 7.27–7.15 (m, 4H, H_aro_); 7.10 (m, 2H, H_aro_); 4.45 (d, *J* = 5.8 Hz, 2H, CH_2_); 3.50 (s, 2H, CH_2_); 2.70 (t, *J* = 6.1 Hz, 2H, CH_2_); 2.45 (t, *J* = 5.6 Hz, 2H, CH_2_); 2.18 (s, 3H, CH_3_). ^13^C NMR (75 MHz, CDCl_3_) δ: 172.5 (CO); 137.5 (C_aro_); 134.8 (C_aro_); 134.3 (C_aro_); 133.7(C_aro_); 131.1 (C_aro_); 129.2 (C_aro_); 129.0 (C_aro_); 128.4 (C_aro_); 127.5 (C_aro_); 127.3 (C_aro_); 62.2 (CH_2_); 53.1 (CH_2_); 41.0 (CH_3_); 40.5 (CH_2_); 32.6 (CH_2_). LCMS *m*/*z* calc for C_18_H_21_Cl_2_N_2_O [M + H]^+^: 351.1, 353.1, 355.1; found: 351.0, 352.9, 355.0 [M + H]^+^.

**3-(Benzylmethylamino)-*N*-(4-cyanobenzyl)propanamide (6):** Compound **6** was synthesized by using 3-chloro*-N*-(4-cyanobenzyl)propanamide **9e** (0.44 mmol, 100 mg) and *N*-benzylmethylamine (4.4 mmol, 579 µL). Purification by thick-layer chromatography (DCM: MeOH(NH_3_), 9.5:0.5, *R*f = 0.3). Yield: 80%. ^1^H NMR (300 MHz, CDCl_3_), δ: 8.90 (br s, 1H, NH); 7.58 (d, *J* = 6.3 Hz, 2H, H_aro_); 7.32 (d, *J* = 8.1 Hz, 2H, H_aro_); 7.21 (m, 3H, H_aro_); 7.07 (m, 2H, H_aro_); 4.43 (d, *J* = 5.9 Hz, 2H, CH_2_); 3.48 (s, 2H, CH_2_); 2.68 (t, *J* = 5.6 Hz, 2H, CH_2_); 2.47 (t, *J* = 5.5 Hz, 2H, CH_2_); 2.21 (s, 3H, CH_3_). ^13^C NMR (75 MHz, CDCl_3_) δ: 172.7 (CO); 144.4 (C_aro_); 137.5 (C_aro_); 132.4 (C_aro_); 129.0 (C_aro_); 128.5 (C_aro_); 128.1 (C_aro_); 127.5 (C_aro_); 118.8 (CN); 110.9 (C_aro_); 62.2 (CH_2_); 52.9 (CH_2_); 42.5 (CH_3_); 41.1 (CH_2_); 32.6 (CH_2_). LCMS *m*/*z* calc for C_19_H_22_N_3_O [M + H]^+^: 308.2; found: 308.2 [M + H]^+^.

**3-(Benzylmethylamino)-*N*-(4-Nitrobenzyl)propanamide (7):** Compound **7** was synthesized by using 3-chloro*-N*-(4-nitrobenzyl)propanamide **9f** (0.41 mmol, 100 mg) and *N*-benzylmethylamine (4.1 mmol, 531 µL). Purification by thick-layer chromatography (DCM: MeOH(NH_3_), 9.5:0.5 (*v*/*v*), *R*f = 0.3). Yield: 37%. ^1^H NMR (300 MHz, CDCl_3_), δ: 9.00 (br s, 1H, NH); 8.15 (d, *J* = 8.7 Hz, 2H, H_aro_); 7.40 (d, *J* = 8.4 Hz, 2H, H_aro_); 7.20 (m, 3H, H_aro_); 7.10 (m, 2H, H_aro_); 4.50 (d, *J* = 6.0 Hz, 2H, CH_2_); 3.50 (s, 2H, CH_2_); 2.70 (t, *J* = 6.2 Hz, 2H, CH_2_); 2.50 (t, *J* = 6.2 Hz, 2H, CH_2_); 2.25 (s, 3H, CH_3_). ^13^C NMR (75 MHz, CDCl_3_) δ: 172.6 (CO); 146.5 (C_aro_); 146.2 (C_aro_); 137.3 (C_aro_); 129.1 (C_aro_); 128.5 (C_aro_); 128.1 (C_aro_); 127.6 (C_aro_); 123.8 (C_aro_); 62.2 (CH_2_); 52.8 (CH_2_); 42.3 (CH_3_); 41.1 (CH_2_); 32.5 (CH_2_); 29.7 (CH_2_). LCMS *m*/*z* calc for C_18_H_22_N_3_O_3_ [M + H]^+^: 328.1; found: 327.9 [M + H]^+^.

### 3.2. Biological Activity

#### 3.2.1. Assay for Binding to Sigma Receptors

Binding assays were performed by Eurofins Cerep (Poitiers, France) according to the protocol of Ganapathy [17]. For the S1R assay, Jurkat cell membranes (10–20 μg protein/tube) were incubated at 37 °C for 2 h in 5 mM Tris-HCl buffer (pH 7.4) with ^3^H-pentazocine (15 nM) and varying concentrations of the test compounds. Under these experimental conditions, the Kd of this radioligand was measured as 16 nM. For the S2R assay, Jurkat cell membranes (10–20 μg protein/tube) were incubated at room temperature for 1 h in the same buffer with [^3^H]-DTG (25 nM) in the presence of (+)-pentazocine (1 μM) to saturate S1R, along with a range of test compound concentrations. Bound radioactivity was quantified by liquid scintillation counting. Nonspecific binding was determined under identical conditions in the presence of 10 μM unlabeled haloperidol. The IC_50_ values (concentration causing a half-maximal inhibition of control-specific binding) and Hill coefficients (nH) were determined by non-linear regression analysis of the competition curves generated with mean replicate values using Hill equation curve-fitting. This analysis was performed using software developed at Cerep (Hill software) and validated by comparison with data generated by the commercial software SigmaPlot ^®^4.0 for Windows^®^ (©1997 by SPSS Inc.). Inhibition constants (*K*i) were calculated from IC_50_ values using the method of Cheng and Prusoff [39].

#### 3.2.2. Assay for Cytotoxicity

Cytotoxicity was assessed in human SH-SY5Y neuroblastoma cells cultured in Dulbecco’s Modified Eagle Medium (DMEM, Gibco, Thermo Fisher Scientific Inc., Waltham, MA, USA) supplemented with 2 mM L-glutamine, 100 μg/mL streptomycin, 100 IU/mL penicillin, 1 mM non-essential amino acids (all from Invitrogen, Thermo Fisher Scientific Inc., Waltham, MA, USA), and 10% (*v*/*v*) heat-inactivated fetal bovine serum (Sigma-Aldrich, Burlington, MA, USA). Cells were maintained at 37 °C in a humidified 5% CO_2_ atmosphere and serum-starved for 24 h to synchronize cultures before treatment with the test compounds at final concentrations of 100, 50, 10, 5, 1, 0.5, 0.1, or 0.05 μM (all in <0.1% DMSO). After 72 h, viability was determined using the MTT-based CellTiter 96^®^ AQueous One Solution Cell Proliferation Assay (MTS, Promega, Madison, WI, USA), following the manufacturer’s protocol. Absorbance was measured at 490 nm, and the results were expressed as a percentage relative to untreated control cells (set at 100%).

#### 3.2.3. Selectivity Profile (Agonist and Antagonist Activities) over a Panel of 58 Recombinant Human Receptors

Rapidly, the tests were performed by Euroscreen (https://euroscreenfast.com/) with recombinant human G-coupled receptor-expressing cell lines using aequorin functional assays for serotonin 5-HT_2A_, 5-HT_2B_, 5-HT_2C_, 5-HT_3_, and 5-HT_5A_, adrenergic α_1a_ and α_1b_, angiotensin AT_1_, cholesistokinin CCFK1, endothelin ETA, histamine H_1_, muscarinic M_1_ and M_3_, neurokinin NK_1_ and NK_2_, vasopressin V_1a_, and vasoactive intestinal peptide VPAC_1_ receptor analysis [20]. Then, cAMP HTRF^TM^ assays were used to analyze serotonin 5-HT_1A_, 5-HT_4E_, 5-HT_6_, and 5-HT_7_, adenosine A_1_, A_2A_, and A_3_, adrenergic β_1_ and β_2_, cannabinoid CB_1_, dopamine D_1_, histamine H_2_, and neuropeptide Y NPY receptors [20]. GTPγS^35^ assays were used to analyze serotonin 5-HT_1B_, adrenergic α_2a_ and α_2b_, dopamine D_2_, muscarinic M_2_ and M_4_, opioid δ, κ and µ, and somatostatine SST4 receptors [21].

Agonist and antagonist activities of the test compound are expressed as the percentage of the activity of the reference agonist at its EC_100_ concentration or the percentage of inhibition of the reference agonist activity at its EC_80_ concentration, respectively.

For the aequorin assays, different cell lines co-expressing the receptors were grown in specific culture media and incubated for 4 h with coelenterazine h. For agonist testing, cell suspensions were mixed with test or reference agonists in a 96-well plate. For antagonist testing, the reference agonist at its EC_80_ (final concentration) was injected to the mix the containing cells and the tested compound. The resulting emission of light was recorded using the Hamamatsu Functional Drug Screening System 6000 [19].

For the cAMP HTRF^TM^ assays, different cell lines co-expressing the receptors were grown in specific culture media and resuspended in assay buffer. For agonist testing on Gs-coupled receptors, the tested compound was added at increasing concentrations. For antagonist testing, the tested compound and the reference agonist at its EC_80_ (final concentration) were added. For agonist and antagonist testing on Gi-coupled receptors, forskolin was added to the reagents listed above. After incubation, cells were lysed and cAMP concentrations were estimated with the HTRF kit, according to the manufacturer’s specifications [20].

For the GTPγS scintillation proximity assay, optimized specific conditions were determined for each receptor. Membrane extracts were prepared from different cell lines co-expressing the receptors, mixed with GDP, and incubated on ice. For agonist testing, the tested compound or the reference agonist, GDP mix, and GTPγ[^35^S] were added. For antagonist testing, the tested compound or the reference antagonist, the reference agonist at historical EC80, GDP mix, and GTPγ[^35^S] were added. GTP concentrations were estimated with a PerkinElmer TopCount reader [21].

#### 3.2.4. In Vitro Evaluation of S1R Functionality

The ability of the compounds to promote dissociation of the S1R–BiP complex was assessed by Amylgen (Montferrier-sur-Lez, France) as described in [22,23]. CHO cells were treated with the test compounds for 30 min at 37 °C, followed by crosslinking with dithiobis(succinimidyl propionate) (DSP; Thermo Scientific, Waltham, MA, USA). Cells were lysed after 15 min on ice, and the lysates were centrifuged. Supernatants were incubated overnight at 4 °C with an anti-S1R antibody (Abcam, Cambridge, UK), and then with Protein A-Sepharose (Invitrogen). After the first centrifugation, pellets were resuspended in radioimmunoprecipitation assay (RIPA) buffer, centrifuged again, and resuspended in 2× sample buffer/bMCE buffer. Following a third centrifugation, the supernatants were analyzed for BiP immunoreactivity using an ELISA kit (SEC343Mu, USCNK Life Sciences, Wuhan, China).

#### 3.2.5. Physicochemical Properties

Standardized in vitro ADME experiments were performed by Eurofins Cerep (Poitiers, France). Solubility assays were performed in PBS at pH 7.4, but also in simulated gastric fluid (SGF) and intestinal fluid (SIF), through the classical shake-flask method described by Lipinski et al., using HPLC-UV/VIS technology [25]. The results were expressed in µM. The compound was classically evaluated at 10 µM for all of the ADME analyses. Bidirectional permeability (apical A to basolateral B pH 6.5/7.4, and basolateral B to apical A pH 7.4/6.5) was assessed in the epithelial colorectal adenocarcinoma cell line (Caco-2 cell line) [26]. The results were expressed as ×10^−6^ cm/s (Papp) and as % recovery. The efflux ratio, i.e., Papp (B→A)/Papp(A→B), was calculated. Plasma protein binding was analyzed by HPLC-MS/MS after 4 h of incubation at 37 °C, as described by Banker et al. [27]. Concentration was evaluated by LC/MS-MS. CYP inhibition was performed in human liver microsomes as described in [28]. Analysis of the respective metabolites was performed by HPLC-MS/MS. Transporter inhibition analysis was also performed in overexpressing cell lines. Respective metabolites for each specific transporter substrate were analyzed by fluorometry or scintillation counting, and % inhibition of control values were calculated [29,30,31,32,33,34,35,36,37,38]. P-glycoprotein transporter substrate assessment was performed by bidirectional permeability analysis (apical A to basolateral B pH 7.4/7.4, and basolateral B to apical A pH 7.4/7.4) in the Caco-2 cell line with and without verapamil, as described in [26]. The results were expressed as ×10^−6^ cm/s (Papp) and as % recovery. The efflux ratio was calculated as Papp(B→A)/Papp(A→B). CYP450 induction was assessed in human hepatocytes by measuring the mRNA levels of seven CYP isoforms via qPCR, as described in [40]. The compound was tested at 1, 10, and 100 μM, and fold-induction values were compared with predefined cut-off thresholds for each isoform and hepatocyte lot.

Plasma stability was evaluated by M2SV (Lille, France) in duplicate. Male CD-1 mouse plasma (lithium-heparinized; Sera Laboratories International Ltd. BioIVT, West Sussex, UK) was pre-incubated for 10 min at 37 °C before addition of the test compound (final concentration 10 μM, 0.1% DMSO). At 0, 60, 1440, and 2880 min, aliquots were transferred to chilled tubes containing acetonitrile and propranolol (1 μM) as internal standard. After mixing and centrifugation (10 min, 10,000 rpm), the supernatants were analyzed, and their degradation half-lives (t_1_/_2_, min and h) were determined by non-linear regression using the Microsoft^®^ XLfit^®^ Excel add-in (IDBS Ltd., Woking, UK).

Cardiotoxicity analysis was performed by Eurofins Cerep (Poitiers, France) as described in [41]. Rapidly, hERG CHO-K1 cells were used to perform automated whole-cell patch-clamp. Compounds were evaluated in duplicate from 0.1 to 10 µM. The degree of inhibition (%) was determined by measuring the tail current amplitude, which was induced by a one-second test pulse to −40 mV after a two-second pulse to +20 mV, before and after drug incubation for 5 min at room temperature. After normalization, activities were expressed as the % inhibition of tail current. Concentration-(log) response curves were fitted to a logistic equation to generate estimates of the 50% inhibitory concentration (IC_50_). The concentration–response relationship of each compound was constructed from the percentage reductions in current amplitude by sequential concentrations.

### 3.3. Molecular Docking

Molecular modeling studies were performed using Gnina 1.3 [42] software, using S1R’s co-crystallized structure (PDB: 5HK2). Briefly, hydrogens and charges were added, and then ligands were separated and used as a reference to generate the searching box, with 4 Å added around this region. Binding poses were analyzed for their binding mode and assessed in comparison with the crystallized analog with the UCSF ChimeraX software, version 1.6.1.

## 4. Conclusions

Central nervous system (CNS) disorders remain among the most critical and challenging therapeutic areas of modern medicine. Accumulating preclinical evidence underscores the promise of selective sigma-1 receptor (S1R) agonists as effective treatments. Building upon our previous demonstration of compound **1** as a safe and efficacious S1R ligand for multiple sclerosis, this study reports the rational pharmacomodulation of this reference scaffold, leading to the design and characterization of six novel derivatives **2**–**7**.

Notably, compound **2**—a simple amide-inverted analog of compound **1**—exhibited a fivefold increase in S1R affinity, S2R/S1R selectivity, and toxicity index, highlighting the impact of subtle structural modifications. Although some moderate off-target affinity toward CNS receptors was observed at 10 µM, compound **2** displayed an overall improved pharmacological profile relative to the reference compound **1**.

In terms of ADME properties, compound **2** demonstrated comparable bioavailability parameters to compound **1**, with slightly enhanced permeability and a modest increase in plasma protein binding. The most significant advancement lies in its markedly improved cardiac safety profile, with an IC_50_ (hERG)/*Ki* (σ_1_) ratio exceeding that of compound **1** by nearly sevenfold.

## 5. Patents

The compounds described in this manuscript have been patented [43].

## Data Availability

The original contributions presented in this study are included in the article/Appendix A. Further inquiries can be directed to the corresponding author.

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
