# Peer review of "Discovery of Novel Benzamide-Based Sigma-1 Receptor Agonists with Enhanced Selectivity and Safety"

_molecules, 2025, doi:10.3390/molecules30173584_

Round 1

Reviewer 1 Report

Comments and Suggestions for Authors

In this research article entitled „Discovery of Novel Benzamide-Based Sigma-1 Receptor Agonists with Enhanced Selectivity and Preclinical Safety for Neurodegenerative Diseases and Brain Ischemia the authors Pascal Carato et al. continue to investigate sigma-1 receptor (S1R) agonists. Six novel benzamide derivatives were designed and synthesized. Among these, compound 2, an amide-inverted analog of previous lead 1 demonstrated superior S1R affinity, improved selectivity over sigma-2 receptor (S2R), and favorable ADME properties. It also demonstrated some moderate off-target affinity toward CNS receptors at 10 μM. The most significant improvement lies in its better cardiac safety profile, with a selectivity ratio exceeding that of compound 1.

This is a very well written and rounded study about agonist/antagonistic activity of novel benzamide analogues against S1R, some of which have potential to enter preclinical studies and I recommend it for publication in Molecules after minor revision. There are some minor issues that have to be addressed in the manuscript. You mentioned that purity of the evaluated compounds was judged to be >96% as determined by UPLC-UV-MS system. In my opinion chromatograms should be added as a proof of compounds’ 2-7 purity since they were tested in vitro. Please add them in the SI file together with peak purity analysis.

Other remarks:

  1. The nomenclature of compounds should be corrected according to the IUPAC Blue Book Guidelines, for example:

N-(4-nitrobenzyl)-3-chloropropanamide (9f) should be named as

3-chloro-N-(4-nitrobenzyl)propanamide (alphabetical order of the substituents)

Another example:

N-(3-bromobenzyl)-3-(benzylmethylamino)propanamide (4) should be

3-(benzylmethylamino)-N-(3-bromobenzyl)propanamide, etc.

Please correct the names in the manuscript and SI file.

  1. Please include Rf values and the corresponding solvents for all prepared compounds that were purified using column chromatography.
  2. Please correct the Lewis structure for NO2 group in structural formula of comp. 7 in SI file
  3. Assignation for CN group from 13C NMR spectrum of comp. 6 is missing in the experimental part. Please add this value.
  4. Some minor spelling and grammar errors throughout the manuscript are listed below, please correct them:

line 90, 96 – N-benzylamine, N-benzylmethylamine – N should be in italics

line 104 – Ki should be in italics –please correct throughout the manuscript (tables and text)

line 107 – prefix para should be written in italics, please correct this for all ortho, meta, para prefixes throughout the manuscript

line 202 – there is a bracket in bold font.

Author Response

In this research article entitled „Discovery of Novel Benzamide-Based Sigma-1 Receptor Agonists with Enhanced Selectivity and Preclinical Safety for Neurodegenerative Diseases and Brain Ischemiathe authors Pascal Carato et al. continue to investigate sigma-1 receptor (S1R) agonists. Six novel benzamide derivatives were designed and synthesized. Among these, compound 2, an amide-inverted analog of previous lead 1 demonstrated superior S1R affinity, improved selectivity over sigma-2 receptor (S2R), and favorable ADME properties. It also demonstrated some moderate off-target affinity toward CNS receptors at 10 μM. The most significant improvement lies in its better cardiac safety profile, with a selectivity ratio exceeding that of compound 1.

This is a very well written and rounded study about agonist/antagonistic activity of novel benzamide analogues against S1R, some of which have potential to enter preclinical studies and I recommend it for publication in Molecules after minor revision.

We thank the reviewer for the positive comments on our work.

There are some minor issues that have to be addressed in the manuscript. You mentioned that purity of the evaluated compounds was judged to be >96% as determined by UPLC-UV-MS system. In my opinion chromatograms should be added as a proof of compounds’ 2-7 purity since they were tested in vitro. Please add them in the SI file together with peak purity analysis.

The compounds presented in this article were synthesized several years ago. Since then, our research team has relocated, and the personnel involved in the study have left the laboratory. As a result, we have been unable to retrieve the original chromatograms. We have reanalyzed the compounds, and their purity now ranges between 80% and 95%. Consequently, we have removed the statement regarding compound purity from the manuscript, as we are unable to provide definitive proof.

Other remarks:

  1. The nomenclature of compounds should be corrected according to the IUPAC Blue Book Guidelines, for example:

N-(4-nitrobenzyl)-3-chloropropanamide (9f) should be named as

3-chloro-N-(4-nitrobenzyl)propanamide (alphabetical order of the substituents)

Another example:

N-(3-bromobenzyl)-3-(benzylmethylamino)propanamide (4) should be

3-(benzylmethylamino)-N-(3-bromobenzyl)propanamide, etc.

Please correct the names in the manuscript and SI file.

The nomenclatures of the compounds have been corrected

  1. Please include Rf values and the corresponding solvents for all prepared compounds that were purified using column chromatography.

The Rf values have been added. We also added the 1H NMR of compound 9b as we realized that it was forgotten in the first manuscript.

  1. Please correct the Lewis structure for NO2 group in structural formula of comp. 7 in SI file

The structure has been corrected

  1. Assignation for CN group from 13C NMR spectrum of comp. 6 is missing in the experimental part. Please add this value.

Corrected. Peak at 118.8 ppm was attributed to CN group

  1. Some minor spelling and grammar errors throughout the manuscript are listed below, please correct them:

line 90, 96 – N-benzylamine, N-benzylmethylamine – N should be in italics

line 104 – Ki should be in italics –please correct throughout the manuscript (tables and text)

line 107 – prefix para should be written in italics, please correct this for all ortho, meta, para prefixes throughout the manuscript

line 202 – there is a bracket in bold font.

The minor errors have been corrected

Reviewer 2 Report

Comments and Suggestions for Authors

The manuscript reports six new benzamide analogs (2–7) derived from a previously published S1R ligand (“compound 1”), with radioligand binding in Jurkat membranes, an S1R–BiP dissociation assay in CHO cells, basic ADME, and a single-point off-target panel. While the topic is within scope, multiple technical and reporting gaps prevent a fair assessment of significance and rigor.

  1. Target engagement is indirect. All pharmacology uses cell membranes or cell assays; no purified S1R or defined recombinant system. Add a direct binding/biophysical assay with recombinant S1R (e.g., DSF/TSA, SPR/BLI) or clearly limit claims. Clarify what your “S2R” assay measures (TMEM97 vs PGRMC1) and its controls.

  2. Thin SAR for a primary med-chem report. Only six analogs with minor ring changes were made and only two advanced to ADME/safety. Expand SAR beyond ring substituents (linker, basic center, scaffold variants) and provide at least one hypothesis-driven optimization cycle.

  3. No in vivo support. If the goal is neuroprotection, include a minimal, mechanism-relevant model or temper all translational language.

  4. Binding section lacks primary data. Káµ¢ values are reported without displacement curves, radioligand Kd, or fit diagnostics. Include raw curves (all replicates), Kd, model/fitting details, and goodness-of-fit.

  5. Implausible dosing/vehicle. Methods state SH-SY5Y exposures up to 100 mM with ≤0.1% DMSO. Correct units (likely µM), document solubility/precipitation in media, specify final vehicle at each dose, and show full viability CRCs.

  6. Numbering/format errors. “Section 5” is missing (jump from 4 → 6). Fix all numbering and unit typos.

  7. Off-target panel is single-point. For targets with ≥50% effect at 10 µM, provide full CRCs with IC50/KB and counterscreens, or soften selectivity claims.

  8. Cardiac safety claims need data depth. Show hERG CRCs (methods, temperature, traces) with n and avoid “selectivity ratios” without PK exposure margins.

This manuscript needs substantially stronger pharmacology, fuller data disclosure, corrected methods/units, and broader SAR before it’s suitable for publication. 

Author Response

The manuscript reports six new benzamide analogs (2–7) derived from a previously published S1R ligand (“compound 1”), with radioligand binding in Jurkat membranes, an S1R–BiP dissociation assay in CHO cells, basic ADME, and a single-point off-target panel. While the topic is within scope, multiple technical and reporting gaps prevent a fair assessment of significance and rigor.

  1. Target engagement is indirect. All pharmacology uses cell membranes or cell assays; no purified S1R or defined recombinant system. Add a direct binding/biophysical assay with recombinant S1R (e.g., DSF/TSA, SPR/BLI) or clearly limit claims. Clarify what your “S2R” assay measures (TMEM97 vs PGRMC1) and its controls.

We aimed to position this work within the context of previous studies. S1R affinity has therefore been carried out using the [3H]-(+)-pentazocine competition binding assay that is still conventionally used by the S1R community to study or compare new compounds with classic prototypical S1R-ligands or team’s own chemical libraries.  In addition to the competition experiment, S1R-targeting drug effect was validated using S1R protein-Bip dissociation assay (Garcia-Pupo L. et al, Acta Pharm Sin B, 2024, 14, 4345). Our goal was to obtain specific, selective and safety S1R-targeting ligands, that is to say compounds with a limited number of off-targets interactions. A second competition binding assay was therefore realized against [3H]-(+)-di-o-tolylguanidine (DTG) binding sites in the presence of excess of non-tritiated (+)-pentazocine which selectively occupies S1R. Historically, [3H]-DTG has been defined as the most useful radioligand in the study of compounds classified as S2R-ligands. S2R was initially proposed as part of the PGRMC1 complex. After several years of controversies, it was reported to be the transmembrane protein TMEM97 also known as MAC30. It was described that PGRMC1 form a complex with TMEM97. However, the competition binding assay does not allow to distinguish TMEM97/S2R, as well as PGRMC1 or DTG residual binding site (RBS) interaction.

Finally, agonist and antagonist activities of our compounds was analyzed across a panel of 58 receptors. These data demonstrate that our compounds do not show any major interaction with CNS off-targets relevant in CNS disorders.

The text has been modified

  1. Thin SAR for a primary med-chem report. Only six analogs with minor ring changes were made and only two advanced to ADME/safety. Expand SAR beyond ring substituents (linker, basic center, scaffold variants) and provide at least one hypothesis-driven optimization cycle.

The pharmacophore of sigma ligands is characterized by two hydrophobic groups and a basic center. In our previous work, we have already explored modifications to the hydrophobic groups, as well as variations in the length and nature of the linker. This manuscript presents the final series of compounds from the project, culminating in a molecule with enhanced properties. The specific objective of the present study was to evaluate the impact of amide bond inversion, as numerous other structural modifications had already been investigated in our earlier studies (Donnier-Maréchal et al, Eur J Med Chem 2017, 964). In the docking section, we added a clarifying sentence to indicate that this analysis did not account for the observed differences in compound affinity.

While reanalyzing the raw data, we discovered an error in the Ki value initially recorded in the lab for compound 7 (110 nM instead of 11 nM). We have corrected this in Table 1 as well as in the corresponding text.

  1. No in vivo support. If the goal is neuroprotection, include a minimal, mechanism-relevant model or temper all translational language.

Since we did not conduct any in vivo experiments on these compounds, we have modified the title of the manuscript. Our initial title had a more translational focus, as the sigma-1 receptor (S1R) is known to be involved in neuroprotection, and we had previously demonstrated such activity for certain S1R ligands developed in our laboratory.

  1. Binding section lacks primary data. Káµ¢ values are reported without displacement curves, radioligand Kd, or fit diagnostics. Include raw curves (all replicates), Kd, model/fitting details, and goodness-of-fit.

We have added the displacement curves as supplementary figures S2 and S3. Details regarding the models used have been included in the Materials and Methods section.

  1. Implausible dosing/vehicle. Methods state SH-SY5Y exposures up to 100 mM with ≤0.1% DMSO. Correct units (likely µM), document solubility/precipitation in media, specify final vehicle at each dose, and show full viability CRCs.

We thank the reviewer for the careful reading of the Materials and Methods section. The maximum exposure dose was indeed 100 µM (not mM). Given that the affinity of our compounds is in the nanomolar range, we evaluated cytotoxicity at 100 µM. As no toxicity was observed at this concentration—approximately 1,000-fold higher than the affinity—we did not perform a full dose–response experiment. Since no cytotoxicity was detected, we reported the value as “> 100 µM” in the table. In the revised version, we have indicated the percentage of cell death observed at 100 µM.

  1. Numbering/format errors. “Section 5” is missing (jump from 4 → 6). Fix all numbering and unit typos.

Corrected

  1. Off-target panel is single-point. For targets with ≥50% effect at 10 µM, provide full CRCs with IC50/KB and counterscreens, or soften selectivity claims.

The compounds showed no detectable affinity for the target in single-concentration assays at 10 µM (performed in duplicate), while their sigma-1 receptor (S1R) affinity is in the nanomolar range—approximately 1,000-fold higher. No concentration–response curves (CRCs) were measured. In the manuscript, we refer to enhanced selectivity, which should be interpreted in light of these limitations.

  1. Cardiac safety claims need data depth. Show hERG CRCs (methods, temperature, traces) with n and avoid “selectivity ratios” without PK exposure margins.

We thank the reviewer for this comment. We realized that the Materials and Methods section was forgotten. It was thus added. In the table 3, the words « selectivity ratio » was erased. We also corrected the text.

This manuscript needs substantially stronger pharmacology, fuller data disclosure, corrected methods/units, and broader SAR before it’s suitable for publication. 

We hope that these corrections adequately address the reviewer’s requests.